# A dietary fatty acid counteracts neuronal mechanical sensitization

Luis O. Romero [ID] [1,2], Rebeca Caires [ID] [1], Alec R. Nickolls [ID] [3,4], Alexander T. Chesler [ID] [3,4✉],
Julio F. Cordero-Morales [ID] [1✉] & Valeria Vásquez [ID] [1✉]

PIEZO2 is the essential transduction channel for touch discrimination, vibration, and proprioception. Mice and humans lacking *Piezo2* experience severe mechanosensory and proprioceptive deficits and fail to develop tactile allodynia. Bradykinin, a proalgesic agent released during inflammation, potentiates PIEZO2 activity. Molecules that decrease PIEZO2 function could reduce heightened touch responses during inflammation. Here, we find that the dietary fatty acid margaric acid (MA) decreases PIEZO2 function in a dose-dependent manner. Chimera analyses demonstrate that the PIEZO2 beam is a key region tuning MA-mediated channel inhibition. MA reduces neuronal action potential firing elicited by mechanical stimuli in mice and rat neurons and counteracts PIEZO2 sensitization by bradykinin. Finally, we demonstrate that this saturated fatty acid decreases PIEZO2 currents in touch neurons derived from human induced pluripotent stem cells. Our findings report on a natural product that inhibits PIEZO2 function and counteracts neuronal mechanical sensitization and reveal a key region for channel inhibition.

[1] 71S. Manassas St. Department of Physiology, College of Medicine, University of Tennessee Health Science Center, Memphis, TN 38103, USA. [2] Integrated Biomedical Sciences Graduate Program, College of Graduate Health Sciences, Memphis, TN 38103, USA. [3] National Center for Complementary and Integrative Health, National Institutes of Health, Bethesda, MD 20892, USA. [4] National Institute of Neurological Disorders and Stroke, National Institutes of Health, Bethesda, MD 20892, USA. ✉email: alexander.chesler@nih.gov; jcordero@uthsc.edu; vvasquez@uthsc.edu

The skin is innervated by sensory neurons expressing mechanosensitive ion channels that allow us to detect and discriminate pleasant from painful touch. The PIEZO2 mechanosensitive ion channel is highly expressed in sensory neurons and Merkel cells where it mediates gentle touch (i.e., brush) and vibration[1–6]. Importantly, research has also shown that PIEZO2 contributes to tactile allodynia, a condition in which innocuous sensations become painful under inflammation[7–10]. Recent findings that *Piezo2*-deficient humans and knockout mice failed to develop sensitization and painful reactions to innocuous touch after skin inflammation suggest that targeting this receptor may be a viable strategy to treating tactile allodynia[9,10].

Mechanosensitive ion channels are known to be modulated by the mechanical properties of the membrane[11–14], intracellular and extracellular proteins[15], and/or cytoskeleton elements[16,17]. There are several lines of evidence, suggesting that PIEZO2 interacts with cellular components to fulfill its physiological role. For instance, PIEZO2's association with stomatin-like protein 3 and cholesterol increases its sensitivity to mechanical stimuli[18,19], sensitization by inflammatory agents via the bradykinin receptor[7], potentiation by Gi-coupled receptor activation[20], and regulation by phosphoinositide lipids[21,22]. Moreover, it has been suggested that PIEZO2 requires cytoskeletal elements such as actin and tubulin for normal function[8]. Together, these data indicate that complex interactions work in concert to tune PIEZO2 function.

We previously explored how fatty acids influence mechanotransduction[23–26]. When enriched in the plasma membrane, the esterified saturated fatty acid margaric acid (MA; C17:0) inhibits closely related PIEZO1 channels by increasing the structural order and stiffness of the membrane, thereby increasing the mechanical threshold required to activate the channel[25]. Given the effect of MA on the mechanical properties of the membrane, we reasoned that MA might also decrease PIEZO2 function. However, unlike PIEZO1 that can be activated by changes in membrane tension alone[27–29], PIEZO2 seems to require an intact cytoskeleton for normal function, as so far it can only be gated in cell-attached or whole-cell patch camp configurations[30]. Therefore, whether MA can efficiently modulate and decrease PIEZO2 activity remains to be determined.

In the current study, we determine that MA decreases PIEZO2 function under both normal and inflammatory-like conditions. We find that MA potently decreases PIEZO2 currents in a wide range of cell types from mice and rats to humans, by increasing the mechanical stimuli needed to activate the channel. Notably, MA supplementation combined with latrunculin A treatment (i.e., a toxin that disrupts actin polymerization), reveal that PIEZO2 mechano-sensitivity relies on both the plasma membrane and the cytoskeletal elements. Analyses of PIEZO chimeras show that the PIEZO2 beam (a large intracellular domain that runs parallel to the membrane and thought to be critical for force sensing[31]) dampens the effect of the membrane on PIEZO2 gating. We determine that in dorsal root ganglia (DRG) neurons, MA efficiently reduces the action potential firing elicited by mechanical stimuli but not by current injection. Importantly, MA decreases PIEZO2 currents potentiated by the proalgesic agent bradykinin, indicating that it might be particularly useful for reducing heightened touch responses during inflammation.

## Results

**MA inhibits PIEZO2 currents in N2A cells**. PIEZO2 channels were first characterized in transfected neuro-2a (N2A) cells using an electrically driven piezo-glass probe[32]. We previously determined that N2A plasma membranes can be enriched with MA after overnight incubation and promote high bending stiffness

and rigidity, as determined by mass spectrometry and atomic force microscopy[25]. Importantly, we found that PIEZO1 displays decreased activity in this membrane environment[25]. To determine whether PIEZO2 can also be modulated by the mechanical properties of the membrane, we transfected *Piezo2* variant V2[33] and measured its mechano-currents after supplementing the N2A$^{Piezo1-/-}$ (i.e., cells in which the *Piezo1* gene has been deleted)[34] cell media with MA, ranging between 1 and 600 μM overnight. We found that MA inhibits PIEZO2 currents in a concentration-dependent manner (Fig. 1a, b and Supplementary Fig. 1a) with an IC$_{50}$ = 190.6 ± 14.7 μM (mean ± SEM; Fig. 1b). Moreover, MA increased by threefold the displacement threshold required to elicit PIEZO2 currents when compared with that of the control cells (Fig. 1c), without affecting the time constant of inactivation (Supplementary Fig. 1b).

MA concentrations higher than 100 μM were required to decrease PIEZO2 currents when using overnight supplementation (Fig. 1b). Fatty acids can accumulate when their consumption is increased through diet[35]. Likewise, we previously demonstrated that MA can also accumulate in the plasma membrane when supplemented in the cell media for several days at low concentrations[25]. Hence, to inhibit PIEZO2 activity with lower doses of MA, we implemented a daily supplementation protocol. Indeed, supplementing N2A$^{Piezo1-/-}$ cells with only 50 μM of MA over the course of 4 days decreased PIEZO2 currents by 65% (Fig. 1d, e). As seen with higher concentrations overnight, our low-dose serial MA supplementation increased the displacement threshold without altering PIEZO2 inactivation (Fig. 1f and Supplementary Fig. 1c). Similar results were obtained when supplementing with 25 μM each day for 8 days (Supplementary Fig. 1d-g), showing that MA concentration could be further reduced if the incubation time was lengthened. Notably, MA also inhibited the activity of two other *Piezo2* variants that are particularly abundant in the trigeminal ganglion (V14 and 16), indicating that it likely affects most alternatively spliced isoforms of this channel[33] (Supplementary Fig. 2a–i). Together, our results demonstrate that MA inhibits PIEZO2 currents by increasing the mechanical threshold required for activation. Thus, as with PIEZO1[25], PIEZO2 is less active after MA increases membrane rigidity.

**PIEZO2 beam tunes MA-mediated channel inhibition**. When comparing PIEZO1 and PIEZO2 activities under increasing MA concentrations, we determined that the IC$_{50}$ for PIEZO1 is 28.3 ± 3.4 μM[25] and for PIEZO2 is 190.6 ± 14.7 μM (mean ± SEM; Fig. 2a). Although there is no direct evidence that PIEZO2 needs an intact cytoskeleton for gating, previous works have shown that it cannot be gated in excised patches, given the notion that the cytoskeleton is required for activation[8,32,34,36]. On the contrary, PIEZO1 can be solely activated by membrane tension in inside-out patches[27–29]. These distinct features may explain why approximately seven times more MA is needed to inhibit PIEZO2 channels than PIEZO1. We previously demonstrated that disrupting the actin filaments does not affect plasma membrane bending stiffness of untreated or MA-enriched N2A cells[25]. Hence, to determine the contribution of the actin cytoskeleton on PIEZO2 gating, we treated MA-enriched cells with latrunculin A and compared their mechanically evoked responses with those cells treated solely with MA. Latrunculin A treatment results in a pronounced leftward shift in the MA dose–response profile for PIEZO2 (IC$_{50}$ = 75.4 ± 13.3 μM; mean ± SEM; Fig. 2b, red circles) that is closer to that of PIEZO1 (Fig. 2a, black triangles). On the other hand, the MA dose–response profile of PIEZO1 is similar in control[25] and latrunculin-treated cells (IC$_{50}$ = 28.3 μM ± 3.4 control vs. 25.6 μM ± 8.4 latrunculin-treated cells, mean ±

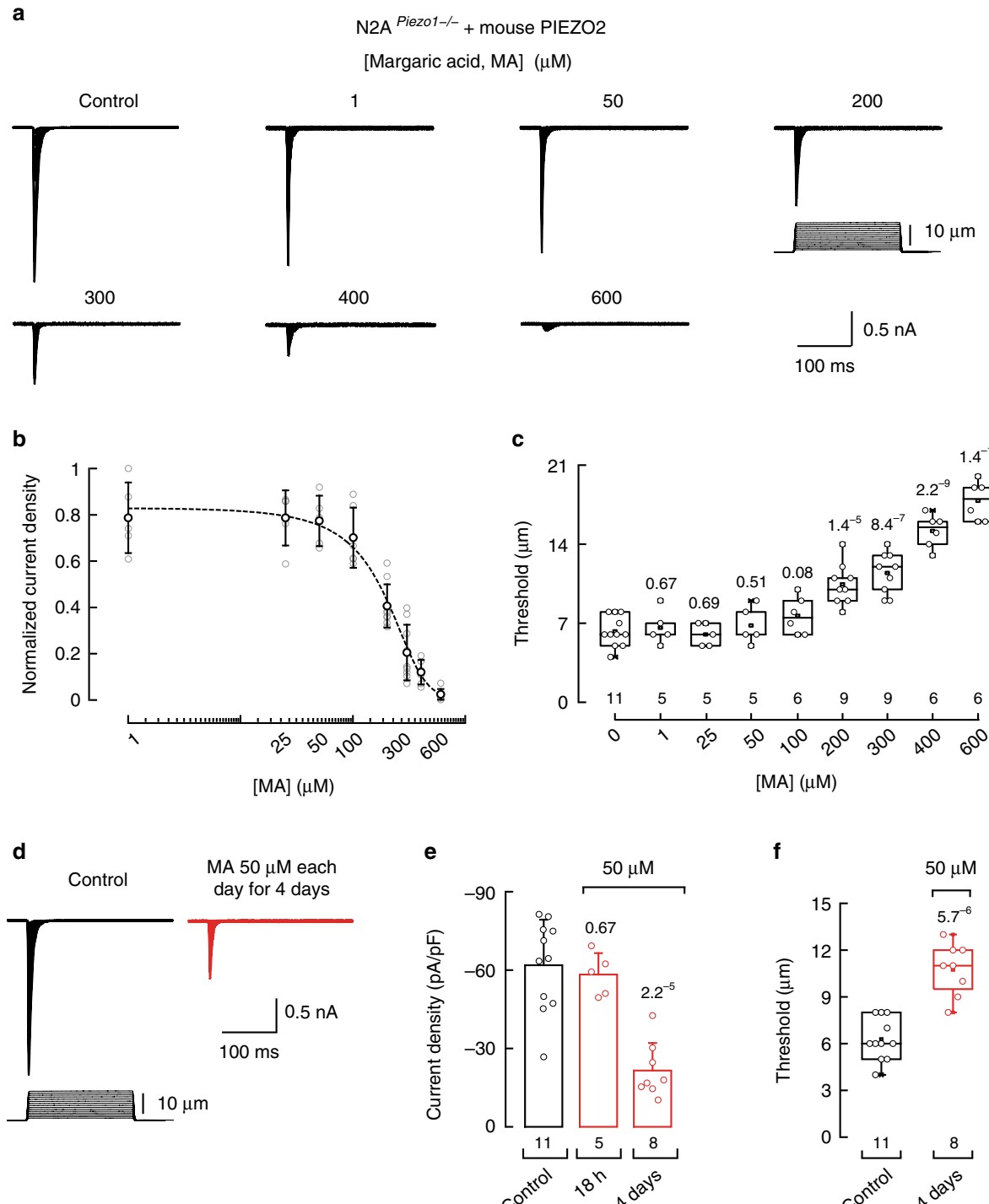

**Fig. 1 MA inhibits heterologously expressed mouse PIEZO2 currents in N2A$^{Piezo1-/-}$ cells. a** Representative whole-cell patch-clamp recordings elicited by mechanical stimulation (at −60 mV) of control and margaric acid (MA)-treated N2A$^{Piezo1-/-}$ cells transfected with *Piezo2* variant 2 (V2). **b** Normalized current densities elicited by maximum displacement of MA-treated N2A$^{Piezo1-/-}$ cells transfected with *Piezo2* V2. A Boltzmann function, Eq. (2), was fitted to the data (IC$_{50}$ = 190.6 ± 14.7 SEM). Circles are mean ± SD. *n* is denoted above the *x*-axis of **c**. **c** Displacement thresholds required to elicit PIEZO2 V2 currents of control and N2A$^{Piezo1-/-}$ cells. Boxplots show mean (square), median (bisecting line), bounds of box (75$^{th}$ to 25$^{th}$ percentiles), outlier range with 1.5 coefficient (whiskers), and minimum and maximum data points. *n* is denoted above the *x*-axis. Two-tailed unpaired *t*-test. **d** Representative PIEZO2 currents (at −60 mV) of control and MA (50 μM each day for 4 days)-treated N2A$^{Piezo1-/-}$ cells transfected with *Piezo2* V2. **e** PIEZO2 V2 current densities elicited by maximum displacement of control and MA (50 μM for 18 h and each day for 4 days)-treated N2A$^{Piezo1-/-}$ cells. Error bars represent SD. *n* is denoted above the *x*-axis. Two-tailed unpaired *t*-test. **f** Displacement thresholds required to elicit PIEZO2 V2 currents of control and MA (50 μM each day for 4 days)-treated N2A$^{Piezo1-/-}$ cells transfected with PIEZO2 V2. Boxplots show mean (square), median (bisecting line), bounds of box (75$^{th}$ to 25$^{th}$ percentiles), outlier range with 1.5 coefficient (whiskers), and minimum and maximum data points. *n* is denoted above the *x*-axis. Two-tailed unpaired *t*-test. *p*-values are denoted above the boxes and bars.

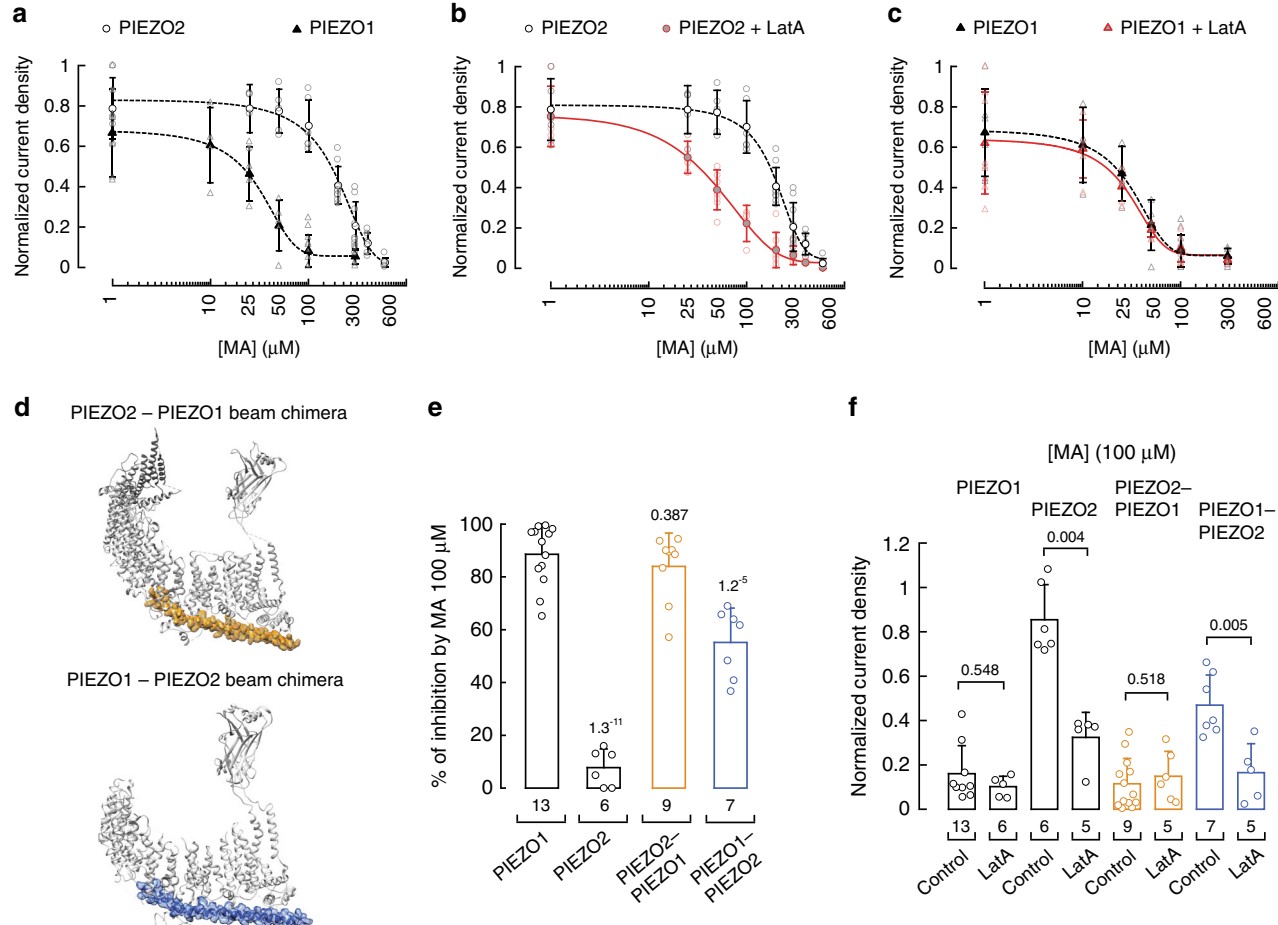

**Fig. 2 Latrunculin A enhances PIEZO2 inhibition by MA. a** Normalized current density elicited by maximum displacement of MA-supplemented N2A cells (expressing endogenous *Piezo1;* triangles) and MA-supplemented N2A$^{Piezo1-/-}$ cells transfected with *Piezo2* (circles). A Boltzmann function, Eq. (2), was fitted to the data. Symbols are mean ± SD. From left to right, *n* for PIEZO1: 6, 4, 4, 5, 13, and 7 and PIEZO2: 5, 5, 5, 6, 9, 9, 6, and 7. **b** Normalized current densities elicited by maximum displacement of MA-supplemented N2APiezo1$^{-/-}$ cells transfected with Piezo2, treated with and without latrunculin A (red and black curves, respectively). A Boltzmann function, Eq. (2), was fitted to the data. Circles are mean ± SD. From left to right, *n* for PIEZO2 with latrunculin A: 6, 5, 7, 5, 8, 7, 4, and 5, and PIEZO2: 5, 5, 5, 6, 9, 9, 6, and 7. **c** Normalized current densities elicited by maximum displacement of MA-supplemented N2A cells (expressing endogenous Piezo1), treated with and without latrunculin A (red and black curves, respectively). A Boltzmann function, Eq. (2), was fitted to the data. Triangles are mean ± SD. From left to right, *n* for PIEZO1 with latrunculin A: 8, 5, 6, 6, 6, and 5, and PIEZO1: 6, 4, 4, 5, 13, and 7. **d** Top, ribbon representation of PIEZO2 monomer (PDB ID: 6KG7; gray) highlighting the residues that were exchanged for those of PIEZO1 (yellow). Bottom, ribbon representation of PIEZO1 monomer (PDB ID: 5Z10; gray) highlighting the residues that were exchanged for those of PIEZO2 (blue). **e** Inhibition by MA supplementation of N2A cells and N2A$^{Piezo1-/-}$ cells transfected with *Piezo2*, and *Piezo2-Piezo1* and *Piezo1-Piezo2* beam chimeras. Error bars represent SD. *n* is denoted above the x-axis. Two-tailed unpaired *t*-test. **f** Normalized current densities elicited by maximum displacement of MA-supplemented N2A cells (expressing endogenous *Piezo1*) and N2A$^{Piezo1-/-}$ cells transfected with *Piezo2*, and *Piezo2-Piezo1* and *Piezo1-Piezo2* beam chimeras treated with and without latrunculin A. Error bars represent SD. *n* is denoted above the x-axis. Two-tailed unpaired *t*-test for PIEZO1 and two-tailed Mann–Whitney test for PIEZO2, and PIEZO2–PIEZO1 and PIEZO1–PIEZO2 beam chimeras. *p*-values are denoted above the bars.

SEM; Fig. 2c), indicating that the mechanism of PIEZO1 current inhibition by MA only depends on the plasma membrane mechanics. Our results further support a previous work that demonstrated that PIEZO1 gating depends on the plasma membrane tension using bleb-attached patches in the absence of the cytoskeleton[28]. Taken together, these results implicate the cytoskeleton as a key determinant of the differential inhibition responses between PIEZO2 and PIEZO1 to MA.

Unlike PIEZO1, the effect that a rigid plasma membrane (i.e., enriched with MA[25]) exerts on PIEZO2 becomes more apparent when the cytoskeleton is pharmacologically disrupted (Fig. 2b, c). We wondered whether modifying PIEZO2 intracellular regions (likely interacting with the cytoskeleton elements) could enhance inhibition by MA, similar to the effect observed with the latrunculin A treatment. Both PIEZO1 and

PIEZO2 contain a 90 Å-long intracellular helix termed the beam (i.e., connects the transmembrane blades with the central pore[36–39]), which, we reasoned, might also tether these channels to the cytoskeleton. Notably, the sequence identity between the PIEZO1 and PIEZO2 beams is low (35%), and thus could account for the different inhibition responses of these channels to MA.

To test this hypothesis, we engineered a PIEZO2 chimera in which we replaced its beam with that of PIEZO1 (Fig. 2d, top). The PIEZO2–PIEZO1 beam chimera displays similar functional properties to PIEZO2, including the reversal potential (7.7 mV PIEZO2 vs. 5.6 mV chimera; Supplementary Fig. 3a, b) and the displacement threshold (6.27 ± 1.35 μm PIEZO2 vs. 6.9 ± 0.8 μm chimera, mean ± SD; Fig. 1c and Supplementary Fig. 3c). However, the time constant of inactivation of this chimera is

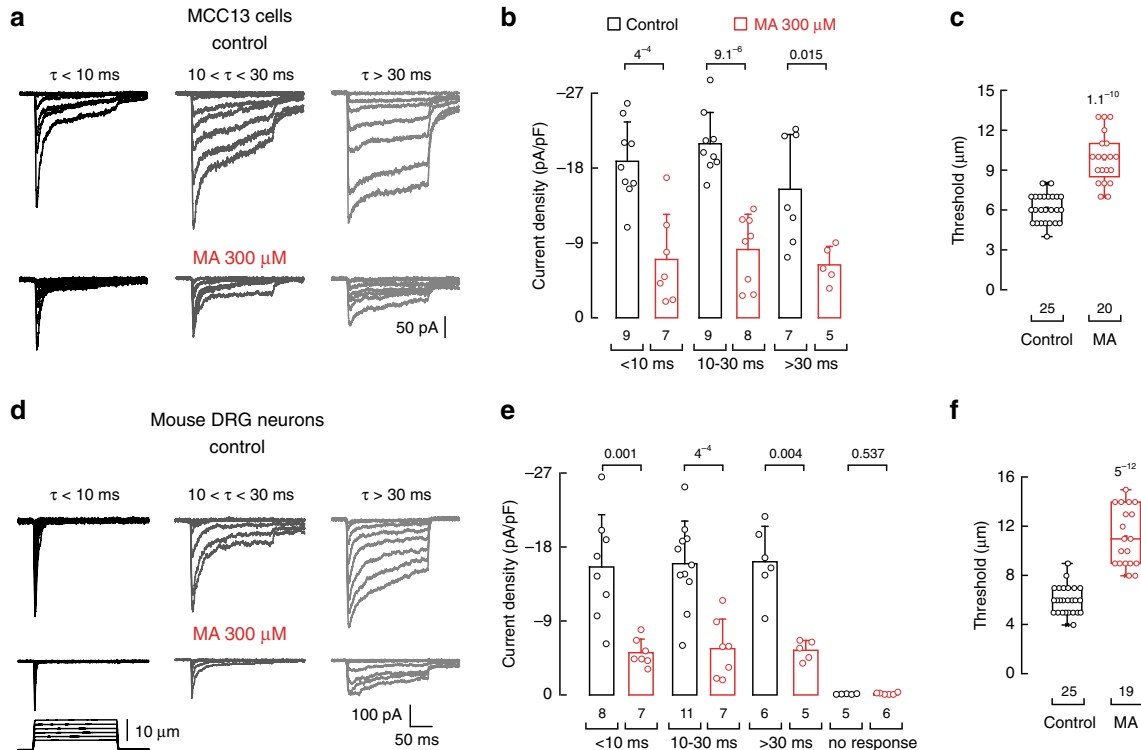

**Fig. 3 MA decreases mechano-activated currents in MCC13 and mouse DRG neurons. a** Representative whole-cell patch-clamp recordings elicited by mechanical stimulation (at −60 mV) of rapidly ($\tau < 10$ ms), intermediate ($10 < \tau < 30$ ms), and slowly inactivating ($\tau > 30$ ms) currents of control (top) and MA (300 μM)-treated (bottom) MCC13. **b** Current densities elicited by maximum displacement of control and MA (300 μM)-treated MCC13 cells. Bars are mean ± SD. *n* is denoted above the *x*-axis. Two-tailed unpaired *t*-test. **c** Displacement thresholds required to elicit mechano-currents of control and MA (300 μM)-treated MCC13 cells. Boxplots show mean (square), median (bisecting line), bounds of box (75th to 25th percentiles), outlier range with 1.5 coefficient (whiskers), and minimum and maximum data points. *n* is denoted above the *x*-axis. Two-tailed unpaired *t*-test. **d** Representative whole-cell patch-clamp recordings elicited by mechanical stimulation (at −60 mV) of rapidly ($\tau < 10$ ms), intermediate ($10 < \tau < 30$ ms), and slowly inactivating ($\tau > 30$ ms) currents of control (top) and MA (300 μM)-treated (bottom) DRG neurons. **e** Current densities elicited by maximum displacement of control and MA (300 μM)-treated DRG neurons. Bars are mean ± SD. *n* is denoted above the *x*-axis. Two-tailed unpaired *t*-test. **f** Displacement thresholds required to elicit mechano-currents of control and MA (300 μM)-treated DRG neurons. Boxplots show mean (square), median (bisecting line), bounds of box (75th to 25th percentiles), outlier range with 1.5 coefficient (whiskers), and minimum and maximum data points. *n* is denoted above the *x*-axis. Two-tailed unpaired *t*-test. *p*-values are denoted above the bars and boxes.

lower than PIEZO2 (7.59 ± 1.96 ms PIEZO2 vs. 1.91 ± 0.46 ms chimera, mean ± SD; Supplementary Fig. 3d). Remarkably, transferring the PIEZO1 beam to PIEZO2 resulted in channels that are much more sensitive to MA (100 μM overnight; Fig. 2e, yellow bar) and this inhibition is not affected by the latrunculin A treatment (Fig. 2f, yellow bars), similar to the results we observed for PIEZO1. These results support that the PIEZO1 beam likely disrupts cytoskeleton regulation of the PIEZO2–PIEZO1 beam chimera.

We also engineered an inverse chimera in which we replaced the PIEZO1 beam with that of PIEZO2 (PIEZO1–PIEZO2 beam chimera; Fig. 2d bottom and Supplementary Fig. 4a). In this case, the beam of PIEZO2 decreased the time constant of inactivation of the chimera (Supplementary Fig. 4b). Future experiments are needed to understand the mechanism by which the beam modulates PIEZO channels inactivation. As expected, the inhibition by MA of the PIEZO1–PIEZO2 beam chimera is not as efficient as observed for PIEZO1 (Fig. 2e, blue bar). Notably, MA inhibition of the PIEZO1–PIEZO2 beam chimera is enhanced by the latrunculin A treatment (Fig. 2f, blue bars), similar to the results we observed for PIEZO2. Our data support the idea that the PIEZO2 beam is a key region tuning MA-mediated channel inhibition. Moreover, both chimeras required a higher mechanical stimulus to open when expressed in cells supplemented with MA (100 μM overnight; Supplementary

Figs. 3c and 4c). However, only the PIEZO1–PIEZO2 beam chimera required a higher mechanical stimulus after latrunculin A treatment (Supplementary Fig. 4c). Taken together, our results highlight that PIEZO2 mechano-sensitivity relies on the synergy between the mechanics of the plasma membrane and interaction with cytoskeleton elements.

**MA decreases mechano-currents and action potential firing.** *Piezo2* is expressed in Merkel cells and its innervating afferents, where it has been shown to transduce skin indentation and whisker deflection into electrical signals[2–4]. In view of the results described above in a heterologous expression system, we asked whether MA could decrease PIEZO2 currents in cells that mediate touch sensation. To this end, we measured the effect of MA on PIEZO2 activity in the human Merkel cell carcinoma cell line (MCC13) and acutely cultured mouse DRG neurons. Similar to dissociated Merkel cells[4], MCC13 displays mechanosensitive currents with a range of inactivation kinetics (Fig. 3a). These mechano-currents have been shown to be mediated by PIEZO2[2,3,40]. As with our experiments using transiently transfected N2A cells, MA supplementation in MCC13 decreased endogenous PIEZO2 currents (Fig. 3a, b) by increasing the displacement threshold (Fig. 3c). Similarly, cultured mouse DRG neurons also exhibit mechano-currents with varying inactivation kinetics. However, in this case, only the rapidly

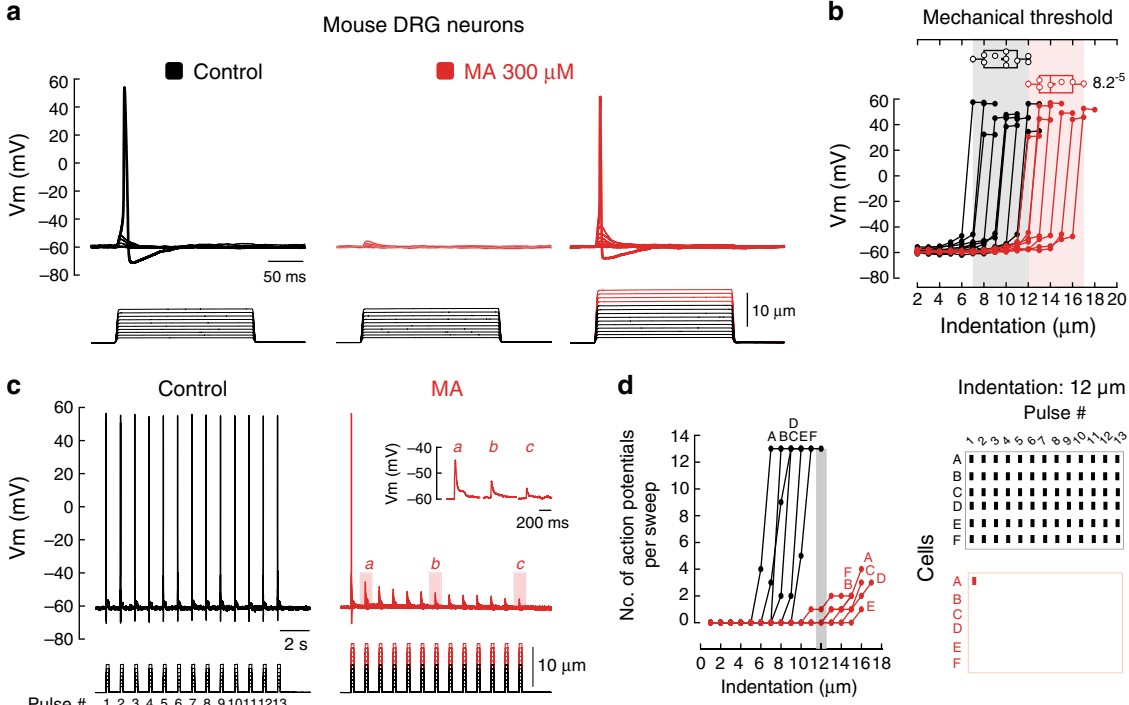

**Fig. 4 MA decreases action potentials elicited by mechanical stimuli in mouse DRG neurons. a** Representative current-clamp recordings of membrane potential changes elicited by mechanical stimulation in control and MA (300 μM)-treated mouse DRG neurons (up to 10 and 15 μm indentation, respectively). **b** Membrane potential peak vs. mechanical indentation of independent control (black; $n = 10$) and MA-treated (red; $n = 7$) mouse DRG neurons. Top panel shows the displacement threshold required to elicit an action potential in these neurons. Boxplots show mean (square), median (bisecting line), bounds of box ($75^{th}$ to $25^{th}$ percentiles), outlier range with 1.5 coefficient (whiskers), and minimum and maximum data points. Two-tailed unpaired *t*-test. *p*-value is denoted on the right. **c** Representative current-clamp recordings of membrane potential changes elicited by a 1 Hz mechanical stimulus train of control (black) and MA-treated (red) mouse DRG neurons. Inset illustrates the progressive decrease in membrane potential as the pulse number progresses. **d** Left: number of action potentials elicited per sweep vs. mechanical indentation of control (black; $n = 6$) and MA-treated (red; $n = 6$) mouse DRG neurons. Right: raster plots displaying the action potentials elicited by 12 μm of indentation. Bars indicate stimuli that elicited action potentials. Columns and rows represent cells and sweeps, respectively.

adapting currents ($\tau < 10$ ms) had been assigned to PIEZO2[32]. Notably, MA supplementation decreased the current magnitude of all DRG neurons mechano-evoked currents, including those known to be mediated by PIEZO2 (Fig. 3d, e) by increasing the displacement threshold (Fig. 3f). These findings translate our heterologous expression results to show that MA decreased the endogenous mechano-currents in diverse cell types known to be involved in mechanosensation.

The detection of touch relies on mechanosensitive ion channels expressed in sensory nerve endings[41]. These channels translate mechanical stimuli into electrical signals, depolarize neurons and, in turn, generate action potentials that propagate toward the central nervous system[42]. PIEZO2 mediates a major proportion of the mechano-activated excitatory currents in mouse DRG neurons[5]. As MA decreases mechano-currents (including those of PIEZO2), we sought to determine whether this saturated fatty acid would also impair the ability of DRG neurons to elicit mechanically activated action potentials. Indeed, we found that MA completely inhibited action potential generation in mouse DRG neurons when indentation steps were smaller than 12 μm (Fig. 4a, left and middle panels). Nevertheless, we were able to elicit action potentials in MA-treated neurons after using larger indentation steps (≥12 μm for MA (red steps) vs. 7–12 μm for control; Fig. 4a right panel and 4b). Furthermore, stimulating DRG neurons with a series of 1 Hz mechanical stimulus trains revealed that cells enriched with MA evoked less action potentials than the control (<12 μm; Fig. 4c), even at large indentation magnitudes (≥12 μm; Fig. 4d). Interestingly, MA-enriched neurons also displayed a progressive decline in

membrane potential as the indentation-pulse number progressed (Fig. 4c, inset). This suggests that after repetitive stimulation, it is more difficult to open mechanosensitive channels in MA-treated neurons.

The mechanically driven depolarization in mouse DRG neurons activates voltage-gated Na$^+$- and K$^+$-channels that are critical for generating action potentials[43]. To determine whether MA impaired the function of ion channels downstream of mechanical activation, we recorded voltage-gated currents in the presence or absence of MA. We found no significant differences in the amplitudes of the voltage-activated inward Na$^+$ and outward K$^+$ currents of control and MA-enriched DRG neurons (Supplementary Fig. 5a-c), as well as the inactivating inward currents after the experimental pulses (Supplementary Fig. 5d). Notably, MA supplementation did not alter membrane potential when measured just after the whole-cell configuration was achieved (Supplementary Fig. 6a). Moreover, we found no differences in the input resistance or the action potential properties (determined by phase plot analysis, as elsewhere[44]) elicited by current injection between control and MA-enriched neurons (Supplementary Fig. 6b-f and 7a-h, and Supplementary Table 1). Our results indicate that MA does not significantly alter DRG neuronal electrical excitability but mainly decreased action potential firing evoked by mechanical stimulation. Nevertheless, the effect of MA on neuronal ion channels should be assessed individually.

Next, we asked whether MA could decrease mechano-currents in rat DRG neurons (Fig. 5a). Mechanical stimulation of rat-cultured DRG neurons elicits currents characteristic of

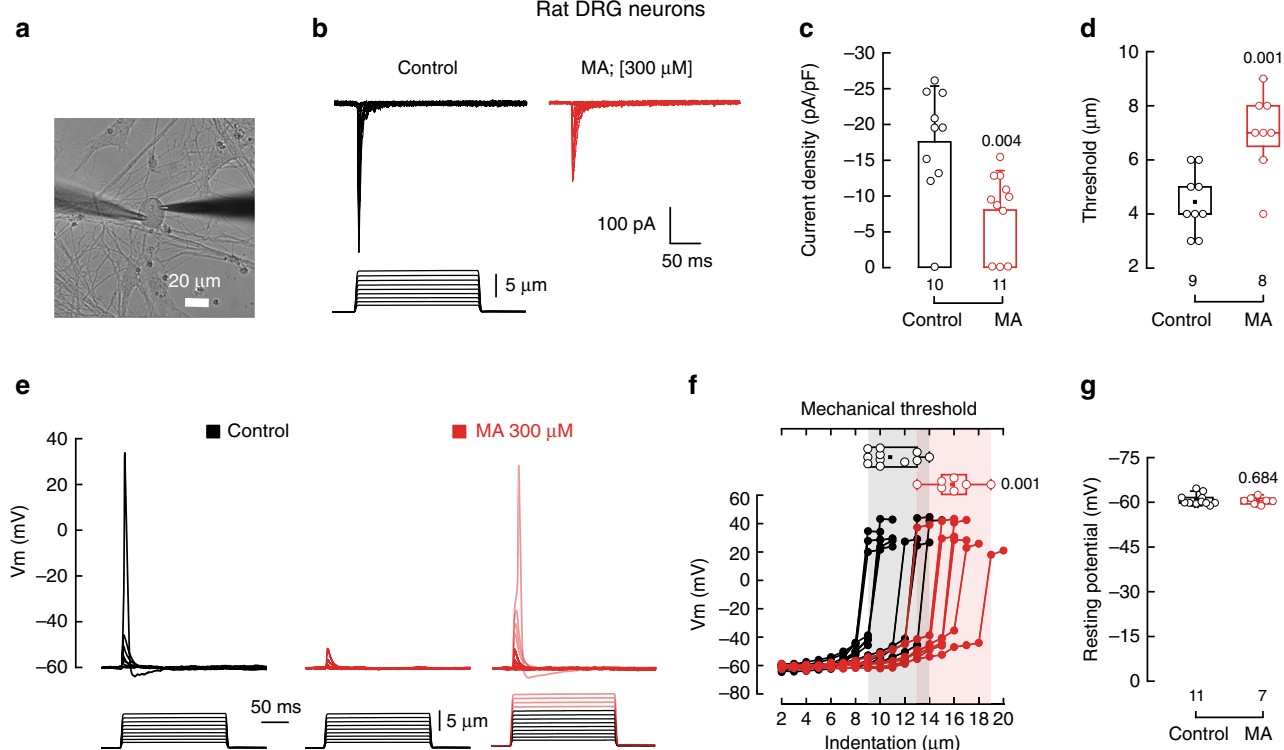

**Fig. 5 MA decreases mechano-activated currents in rat DRG neurons. a** Micrograph showing a rat DRG neuron in the whole-cell patch-clamp configuration ready for mechanical stimulation. Micrograph is representative of at least 12 independent preparations. **b** Representative whole-cell patch-clamp traces of mechanically activated currents of control and MA (300 µM)-treated rat DRG neurons. **c** Current densities elicited by maximum displacement of control and MA (300 µM)-treated rat DRG neurons. Bars are mean ± SD. $n$ is denoted above the $x$-axis. Two-tailed unpaired $t$-test. **d** Boxplots show mean, median, and the 75th to 25th percentiles of the displacement thresholds required to elicit currents of control and MA (300 µM)-treated rat DRG neurons. $n$ is denoted above the $x$-axis. Two-tailed unpaired $t$-test. **e** Representative current-clamp recordings of membrane potential changes elicited by mechanical stimulation in control and MA (300 µM)-treated rat DRG neurons (up to 9 and 13 µm indentation, respectively). **f** Membrane potential peak vs. mechanical indentation of independent control (black; $n = 11$) and MA-treated (red; $n = 7$) rat DRG neurons. Top panel shows the displacement threshold required to elicit an action potential in these neurons. Boxplots show mean (square), median (bisecting line), bounds of box (75th to 25th percentiles), outlier range with 1.5 coefficient (whiskers), and minimum and maximum data points. Two-tailed Mann–Whitney test. **g** Membrane resting potential values recorded briefly after whole-cell current-clamp configuration was achieved from control and MA (300 µM)-treated rat DRG neurons. Error bars represent SD. $n$ is denoted above the $x$-axis. Two-sided permutation $t$-test. $p$-values are denoted above the bars and right of the box.

PIEZO2 channels (Fig. 5b left panel and Supplementary Fig. 8a). As determined for mouse DRG neurons, MA supplementation decreased PIEZO2 currents (Fig. 5b, c) by increasing the displacement threshold (Fig. 5d). Unlike mice, MA supplementation decreases PIEZO2 inactivation (Supplementary Fig. 8b). The reason behind this difference remains to be determined. We also tested the ability of MA to impair mechanically activated action potentials in rat DRG neurons. As expected, MA inhibited action potential generation when indentation steps were smaller than 13 µm (Fig. 5e, f). Nevertheless, we were able to elicit action potentials in MA-treated neurons after using larger indentation steps (≥13 µm for MA (red steps) vs. 9–14 µm for control; Fig. 5f). Stimulating rat DRG neurons with a series of 1 Hz mechanical stimulus trains revealed that neurons enriched with MA evoked less action potentials (Supplementary Fig. 8c, d). Notably, MA did not significantly alter the resting membrane potential (Fig. 5g) and electrical excitability of these rat neurons (Supplementary Fig. 8e-g). Taken together, MA decreased mechano-currents and mechanical excitability of mouse and rat DRG neurons.

**MA counteracts PIEZO2 bradykinin sensitization.** Tissue damage is frequently accompanied by the accumulation of

proalgesic inflammatory agents such as bradykinin, eicosanoids, and protons[43]. These inflammatory molecules bind or interact with diverse membrane proteins, activate intracellular signaling cascades, and increase sensitivity to sensory stimuli leading to allodynia or hyperalgesia[43]. Dubin and colleagues demonstrated that PIEZO2 mechanically evoked currents are potentiated downstream of the activation of the bradykinin beta 2 receptor in mouse DRG neurons[7]. Molecules that decrease PIEZO2 sensitization could therefore be beneficial in treating mechanical allodynia. Given that MA significantly decreased PIEZO2 currents in DRG neurons (Fig. 3d-f), we wondered whether MA supplementation also decreased bradykinin-mediated PIEZO2 sensitization. Similar to previous reports[7], we found that acute bradykinin perfusion sensitized mechano-activated currents of mouse DRG neurons (2.5-fold increase; Fig. 6a top and 6b, and Supplementary Fig. 9 black bar). As predicted, MA supplementation decreased the mechano-currents, even after sensitization with bradykinin (Fig. 6a bottom). Remarkably, the currents recorded in MA-supplemented neurons after bradykinin administration (Fig. 6a and Supplementary Fig. 9) closely resembled those of control DRG neurons (3.36 ± 1.68 pA/pF control vs. 3.64 ± 1.96 pA/pF bradykinin with MA, mean ± SD; Fig. 6b). Hence, MA reduced the mechano-currents to non-inflammatory-like levels. Similar findings were observed

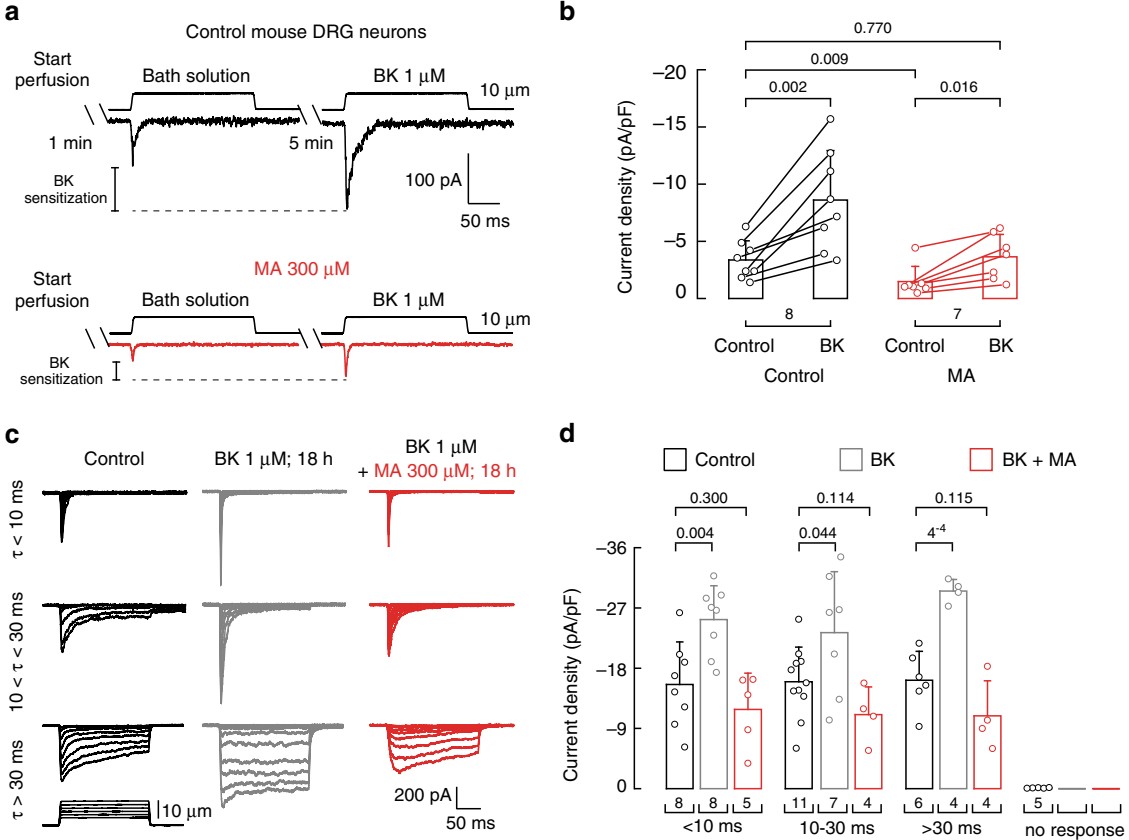

**Fig. 6 MA recovers normal mechanical response in sensitized mouse DRG neurons. a** Representative whole-cell patch-clamp traces of mechanically activated currents after perfusing bath solution (60 s) and bath solution containing bradykinin (BK; 300 s; 1 μM) consecutively to control (black) and MA (300 μM)-treated mouse DRG neurons. **b** Current densities elicited by 10 μm displacement of control and MA (300 μM)-treated mouse DRG neurons perfused for with bath solution (60 s) and with bath solution containing bradykinin (BK; 300 s, 1 μM) consecutively. Bars are mean ± SD. Data samples are paired. $n$ is denoted above the x-axis. Two-tailed paired $t$-test for untreated control before and after BK, two-tailed unpaired $t$-test for untreated control and MA-treated neurons after BK, two-tailed Mann–Whitney for untreated control and MA-treated control before BK, and two-tailed Wilcoxon matched-pairs signed-ranks test for MA-treated neurons before and after BK. **c** Representative whole-cell patch-clamp recordings elicited by mechanical stimulation (at −60 mV) of rapidly ($\tau < 10$ ms), intermediate ($10 < \tau < 30$ ms), and slowly inactivating ($\tau > 30$ ms) currents of control (black), BK (gray; 1 μM for 18 h) and BK + MA (red; 1 μM and 300 μM, respectively, for 18 h)-treated mouse DRG neurons. **d** Current densities elicited by maximum displacement of mechanically activated currents elicited by mechanical stimulation (at −60 mV) of rapidly ($\tau < 10$ ms), intermediate ($10 < \tau < 30$ ms), and slowly inactivating ($\tau > 30$ ms) currents of control, BK (1 μM; 18 h) and BK + MA (1 μM and 300 μM, respectively; 18 h both)-treated mouse DRG neurons. Bars are mean ± SD. Two-tailed unpaired $t$-test. $n$ is denoted above the x-axis. p-values are denoted above the bars.

with longer exposures to bradykinin. Overnight incubation with bradykinin potentiated the magnitude of all mechano-evoked currents of the DRG neurons (Fig. 6c middle panel and 6d). Notably, combined overnight incubation with bradykinin and MA restored the current densities to those of control neurons (Fig. 6c, d). Altogether, these results demonstrate that enriching the plasma membrane with MA counteracted the mechanical sensitization evoked by bradykinin.

**MA decreases mechano-currents in human iPSC-derived neurons.** Our previous results demonstrated that enriching the plasma membrane with MA had an inhibitory effect on murine PIEZO2 function in vitro and ex vivo. Hence, we further tested the effect of MA on human sensory neurons. Recently, we developed a platform to robustly and reproducibly reprogram human induced pluripotent stem cells (iPSCs) into well-characterized neurons that have functional and transcriptional hallmarks indicative of low threshold mechano-receptors (Fig. 7a and ref. [45]). Notably, these in vitro-derived touch neurons, all have mechanically evoked currents that are entirely dependent on PIEZO2 expression[45]. Overnight incubation of human iPSCs with

MA (300 and 600 μM) significantly reduced endogenous PIEZO2 currents (Fig. 7b, c). Furthermore, we confirmed these results by measuring mechano-currents of human PIEZO2 transfected in N2A$^{Piezo1-/-}$ cells supplemented with MA (Supplementary Fig. 10a, b). Moreover, supplementing human iPSC-derived neurons with 50 μM MA each day for 5 days also significantly decreased PIEZO2 currents (Fig. 7b, c). As expected, MA increased the mechanical threshold needed to activate the human channel without altering the time constant of inactivation, mirroring the results obtained with the murines ortholog (Fig. 7d, e and Supplementary Fig. 10c, d). Similar to mouse and rat-cultured DRG neurons, MA did not significantly change voltage-activated inward Na$^+$ and outward K$^+$ currents when compared with control human iPSC-derived neurons (Fig. 7f–h). These findings indicate that MA mainly affected mechanically activated currents without significantly affecting the electrical excitability of human sensory neurons.

## Discussion

Mechanosensory ion channels are essential as they allow us to detect innocuous, pleasurable, alarming, or painful stimuli[46].

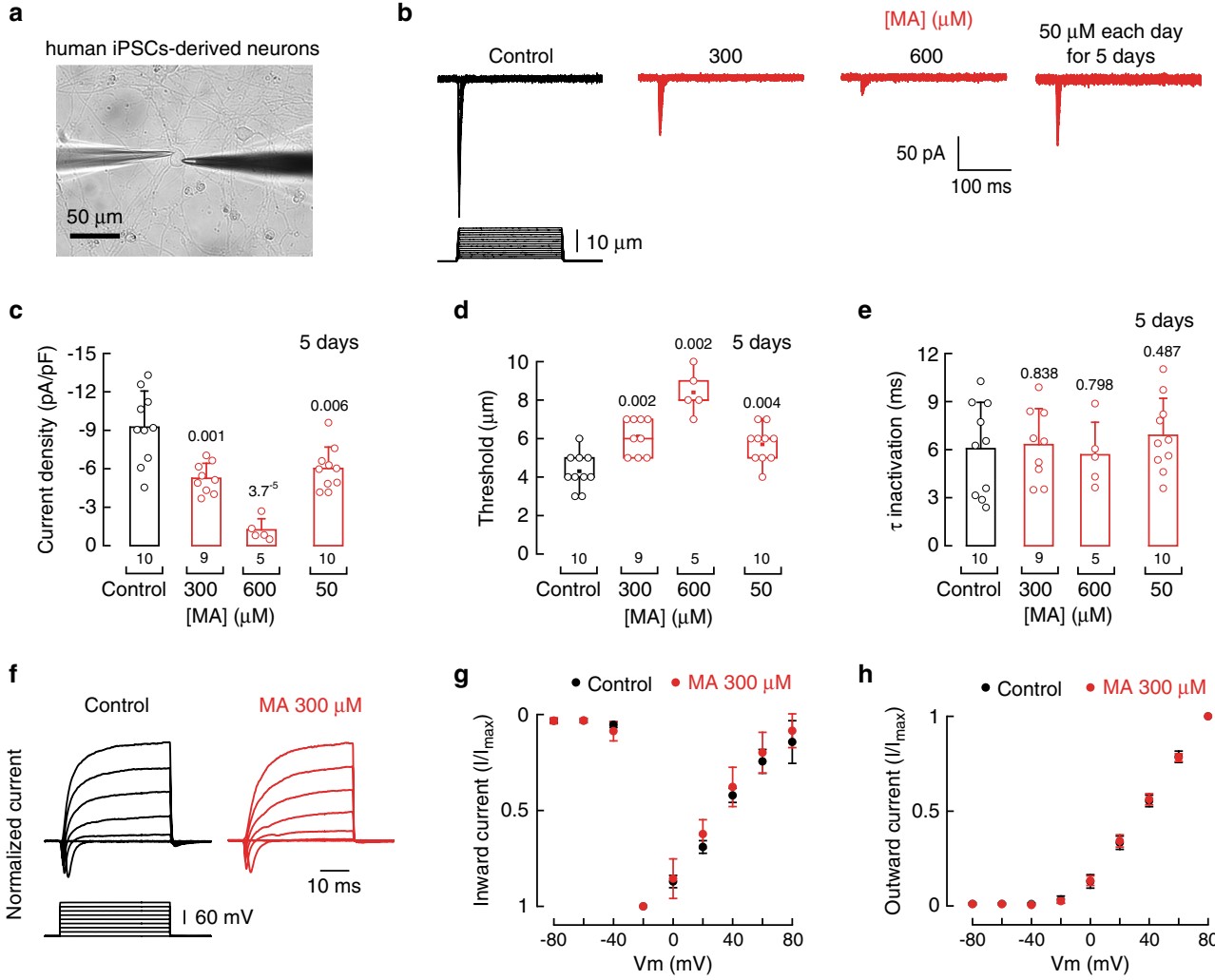

**Fig. 7 MA decreases mechano-activated currents in human iPSCs-derived neurons. a** Micrograph of a human iPSCs-derived neuron in the whole-cell patch-clamp configuration ready for mechanical stimulation. Micrograph is representative of at least 10 independent preparations. **b** Representative whole-cell patch-clamp traces of mechanically activated currents of control and MA (300 and 600 μM for 18 h and 50 μM each day for 5 days)-treated iPSC-derived neurons. **c** Current densities elicited by maximum displacement of control and MA (300 and 600 μM for 18 h and 50 μM each day for 5 days)-treated iPSC-derived neurons. Bars are mean ± SD. n is denoted above the x-axis. Two-tailed unpaired t-test. **d** Displacement thresholds required to elicit currents of control and MA (300 and 600 μM for 18 h and 50 μM each day for 5 days)-treated iPSC-derived neurons. Boxplots show mean (square), median (bisecting line), bounds of box (75th to 25th percentiles), outlier range with 1.5 coefficient (whiskers), and minimum and maximum data points. n is denoted above the x-axis. Two-tailed Mann–Whitney test for control and 300 μM MA and control and 600 μM MA, and two-tailed unpaired t-test for control and 50 μM for 5 days. **e** PIEZO2 time constant of inactivation elicited by maximum displacement of control and MA (300 and 600 μM for 18 h and 50 μM each day for 5 days)-treated iPSC-derived neurons. Bars are mean ± SD. n is denoted above the x-axis. Two-tailed unpaired t-test. **f** Representative whole-cell patch-clamp recordings of control and MA (300 μM)-treated iPSC-derived neurons depolarized in a stepwise manner from a membrane potential of −80 mV. **g** Normalized inward current densities elicited by stepwise depolarization from −80 mV of control and MA (300 μM)-treated DRG neurons. Circles are mean ± SD. n = 8. **h** Normalized outward current densities elicited by stepwise depolarization from −80 mV of control and MA (300 μM)-treated DRG neurons. Circles are mean ± SD. n = 8. p-values are denoted above the bars and boxes.

PIEZO2 has emerged as the principle molecular detector for specific aspects of gentle touch (vibration sensing and tactile discrimination) via its expression in specialized epithelial cells (Merkel cells) and peripheral sensory neurons[2–5,46–48]. Importantly, PIEZO2 is also essential for the experience of touch-evoked pain after injury or under chronic inflammation[9,10], a common condition known as tactile allodynia that remains difficult to treat[46]. Specifically, proalgesic agents (such as bradykinin) produced in response to tissue injury potentiate PIEZO2 response[7,8]. In the ideal scenario, new treatment approaches will be developed to specifically counteract this type of pain without

impairing normal touch function. In the current study, we demonstrated that application of MA, a natural product found in several sources of food such as dairy and mutton fat, rye, and fish[49,50], effectively reduces PIEZO2 function.

Previously, we showed that MA could be efficiently enriched in various cell types and, as a consequence, alter the activation profile of PIEZO1 channels by increasing the plasma membrane structural order and rigidity[25]. Interestingly, previous works suggests that PIEZO1 and PIEZO2 have distinct gating mechanisms[30,51]. For instance, Cox et al.[28] demonstrated that PIEZO1 is solely gated by bilayer tension in the absence of the

cytoskeleton, whereas current evidence suggests that PIEZO2 gating might depend on the cytoskeleton[8]. Whether PIEZO2 activation also relies on the mechanical properties of the plasma membrane is less understood. Our data indicate that PIEZO2 activation is impaired by membrane rigidity and that the beam domain and the cytoskeleton counteracted the effect of the membrane tension. This highlights that the beam is a key region for tuning MA-mediated channel inhibition. Putting our data in a larger context, we favor the idea that PIEZO2 works as part of a force-bearing center[52]. In this model, PIEZO2 function is tightly controlled by a platform comprising the plasma membrane (fatty acid tails and polar head groups), stomatin-like proteins[18,19], cytoskeleton elements (actin and microtubules[8]), and extra-cellular tethers (e.g., focal adhesions[53]).

How selective is MA for mechano-sensing? Our data show that MA reduced the ability of DRG neurons to fire action potentials upon mechanical stimulation without significantly affecting the membrane potential, input resistance, current-elicited action potentials properties, and voltage-activated inward $Na^+$ and outward $K^+$ currents. As such, MA seems to have many properties preferable over other identified mechanoreceptor antagonists such as the tarantula peptide GsMTx-4[54,55], which failed to inhibit mechano-currents from mouse DRG neurons[56]. In addition, it has been shown that the conopeptide analog NMB-1 only inhibits the intermediate and slowly inactivating mechanosensitive currents, but not the rapid ones[56]. Importantly, unlike peptide toxins, MA is able to inhibit all mechanosensitive currents of mouse and rat DRG neurons and of human iPSCs-derived neurons.

With the recent discovery that PIEZO2 is required for tactile allodynia in mice and humans[9,10], this channel has emerged as a promising target to attenuate mechanical pain during inflammatory conditions. Our data show that MA is able to counteract PIEZO2 sensitization by bradykinin, by reducing the mechano-currents to non-inflammatory-like levels. In-vivo experiments are needed to determine MA's efficacy in reducing the heightened touch responses during inflammation when used as a dietary supplement or topical ointment.

## Methods

**Ethics approval.** Mice procedures described below were reviewed and approved by the University of Tennessee Health Science Center Institutional Animal Care and Use Committee. All methods were carried out in accordance with the approved guidelines. The iPSC line was derived and characterized previously[57]. Written informed consent for patient skin biopsies was obtained by a qualified investigator (protocol 12-N-0095 approved by the National Institute of Neurological Disorders and Stroke, National Institutes of Health).

**Cell culture.** *Piezo1* knockout mouse N2A (N2A$^{Piezo1-/-}$) cells were a gift from Dr. Gary R. Lewin. N2A$^{Piezo1-/-}$ cells were cultured in Dulbecco's Modified Eagle Medium (DMEM), 5% penicillin–streptomycin, and 10 % fetal bovine serum (FBS). Human Merkel cell carcinoma cell line (MCC13 cells; Cell Bank Australia reference number: CBA1338) were obtained from Sigma and were used according to the manufacturer's protocol. MCC13 cells were cultured in RPMI 1640 (with 2 mM L-glutamine + 25 mM HEPES; Sigma), 5% penicillin–streptomycin, and 10% FBS, and DRG neurons were cultured in DMEM, 1% penicillin–streptomycin, 1% MEM vitamin solution, 1% L-glutamine, and 10% horse serum. Prior to electrophysiological measurements, N2A$^{Piezo1-/-}$, MCC13, and DRG neurons were supplemented overnight ($\approx$18 h) with MA, unless otherwise stated. For accumulation assays, cells were supplemented with 50 µM MA every 24 h for 5 days. MA was obtained from Nu-Chek Prep, Inc. The cultured cells were maintained at 37 °C, 95% relative humidity, and 5% $CO_2$. N2A$^{Piezo1-/-}$ cells were co-transfected with 75–200 ng ml$^{-1}$ of mmPiezo2 variants (2, 14, and 16), 500 ng ml$^{-1}$ of the Piezos beam chimeras, 200 ng ml$^{-1}$ hPiezo2 cloned in pcDNA3.1, and 50 ng ml$^{-1}$ GFP-pMO; using Lipofectamine 2000 (Thermo Fisher Scientific) according to the manufacturer's instructions, and recorded 48 h later. Fatty acids were supplemented 18–24 h prior to recording, unless stated otherwise.

Primary cultures of mouse DRG neurons were obtained from 8–12-week-old male C57BL/6 mice. Mice were anesthetized with isoflurane and then killed by cervical dislocation. DRGs were dissected and kept on ice in Hank's balanced salt solution 1 × (HBSS without CaCl$_2$ and MgCl$_2$). Then DRGs were incubated in 1

mg/mL collagenase B (Sigma) in HBSS, at 37 °C and 5% $CO_2$ and, after 1 h, were dissociated in medium without serum. The cell suspension solution was centrifuged for 8 min at 800 r.p.m. The obtained pellet was resuspended in DMEM complete media containing 1% penicillin–streptomycin, 1% MEM vitamin solution, 1% L-glutamine, and 10% horse serum. Cells were cultured on coverslips pretreated with poly-L-lysine. All cultured neurons were used after 18–24 h. Rat DRG neurons (R8820N-10) were obtained from Cell Applications, Inc. Neurons were thawed and cultured according to the manufacturer's protocol and were used between days 3 and 5 after thawing.

**Human iPSC-derived neurons.** For generating human peripheral sensory neuron cultures, a version of the healthy control WTC11 iPSC line was used. This line was previously engineered to harbor a doxycycline-inducible NGN2-BRN3A construct that enables rapid and efficient sensory neuron differentiation[45]. Undifferentiated iPSCs were maintained in E8 flex medium (Invitrogen) on polystyrene plates coated with Matrigel (Corning). The medium was exchanged every 1–3 days and the cells were passaged every 4–7 days with Accutase (Invitrogen) and plated overnight with 10 µM of the ROCK-inhibitor Y-27632 (Tocris). For sensory neuron differentiation, iPSCs were seeded at 20,000 cells × (cm$^2$)$^{-1}$ in neural differentiation medium (NDM) on Matrigel-coated plates. The cells were then re-plated after 48 h at 50,000 cells × (cm$^2$)$^{-1}$ onto dishes coated with polyethyleneimine (Sigma-Aldrich) and laminin (Invitrogen). NDM consisted of 1 : 1 DMEM/F12 and Neurobasal medium supplemented with N2, B27, and GlutaMAX (all from Invitrogen) at manufacturer-recommended dilution. Doxycycline (2 µg × ml$^{-1}$) (Clontech) was included in the medium for the duration of the culture. Y-27632 (10 µM) was supplemented for the first 48 h and the following neurotrophic factors were added from day 8 onward at 10 ng/ml each: BDNF, GDNF, β-NGF, and NT-3 (all from R&D systems). Full medium changes were made every other day until after day 8 and then half volume medium changes were done every other day for the remaining time in culture. Before electrophysiological recording, a subset of dishes was supplemented with 300 or 600 µM for 18 h or 50 µM for 5 days of MA. All recordings were performed on neurons cultured for 14–16 days.

**Electrophysiology.** For whole-cell recordings, the bath solution contained 140 mM NaCl, 6 mM KCl, 2 mM CaCl$_2$, 1 mM MgCl$_2$, 10 mM glucose, and 10 mM HEPES (pH 7.4; 300 mOsm). The pipette solution for voltage-clamp recordings of mechano-currents contained 140 mM CsCl, 5 mM EGTA, 1 mM CaCl$_2$, 1 mM MgCl$_2$, and 10 mM HEPES (pH 7.2); for current-clamp and voltage-clamp recordings of voltage-dependent currents, 140 mM KCl, 6 mM NaCl, 2 mM CaCl$_2$, 1 mM MgCl$_2$, 10 mM glucose, and 10 mM HEPES (pH 7.4; 300 mOsm). MA and bradykinin acetate salt (Sigma) perfused during experiments were dissolved in the bath solution to a final concentration of 300 µM for 2 min, and 1 µM for 5 min respectively; for long exposure experiments, bradykinin was supplemented to the culture medium and added to the cells 18–24 h prior recording. For cytoskeleton disruption experiments, N2A$^{Piezo1-/-}$ cells were incubated in media supplemented with 1 µM latrunculin A (Cayman Chemicals) for 1 h prior recordings. Pipettes were made out of borosilicate glass (Sutter Instruments) and were fire-polished before use until a resistance between 3 and 5 MΩ was reached.

During mechanical stimulation, currents were recorded at a constant voltage (−60 mV, voltage-clamp unless otherwise noted) and voltages were recorded without injecting current (current-clamp). Both variables were sampled at 100 kHz and low-pass filtered at 10 kHz using a MultiClamp 700B amplifier and Clampex (Molecular Devices, LLC). To measure voltage-dependent currents, a square-pulse protocol consisting of 40 ms 20 mV incremental steps starting from −80 mV in 500 ms intervals with P/4 subtraction was used. To record action potentials evoked by current injection, 40 ms 20 pA incremental steps were injected in 500 ms intervals. In both cases, variables were sampled at 20 kHz and low-pass filtered at 10 kHz. Leak currents before mechanical stimulations were subtracted offline from the current traces and data were digitally filtered at 2 kHz with ClampFit (Molecular Devices, LLC). Recordings with leak currents >200 pA, with access resistance >10 MΩ, and cells which giga-seals did not withstand at least six consecutive steps of mechanical stimulation were excluded from analyses.

**Mechanical stimulation.** For indentation assays, N2A$^{Piezo1-/-}$, MCC13 cells, DRG neurons, and human iPSC-derived neurons were mechanically stimulated with a heat-polished blunt glass pipette (3–4 µm) driven by a piezo servo controller (E625, Physik Instrumente). The blunt pipette was mounted on a micromanipulator at an ~45° angle and positioned 3–4 µm above from the cells without indenting them. Displacement measurements were obtained with a square-pulse protocol consisting of 1 µm incremental indentation steps, each lasting 200 ms with a 2 ms ramp in 10 s intervals. The threshold of mechano-activated currents for each experiment was defined as the indentation step that evoked the first current deflection from the baseline. For current-clamp experiments, the mechanical threshold was defined as the indentation step that evoked the first action potential.

For pulse train assays, 13 s sweeps with a train rate of 1 Hz of square pulses lasting 200 ms were used. Subsequent sweeps had increments of 1 µm. Only cells that did not detach throughout stimulation protocols were included in the analysis. The piezo servo controller was automated using a MultiClamp 700B amplifier through Clampex (Molecular Devices, LLC).

**Data analysis**. Results were expressed as means ± SD (unless otherwise noted). All boxplots show mean (square), median (bisecting line), bounds of box (75th to 25th percentiles), outlier range with 1.5 coefficient (whiskers), and all data points including maximum and minimum. Data were plotted using OriginPro (from OriginLab) and Estimation Stats[58]. The time constant of inactivation τ was obtained by fitting a single exponential function, Eq. (1), between the peak value of the current and the end of the stimulus:

$$f_{(t)} = \sum_{i=1}^{n} A_i^* e^{-t/\tau_i} + C \tag{1}$$

where $A$ = amplitude, $\tau$ = time constant, and the constant $y$-offset $C$ for each component $i$. Sigmoidal fitting was done using OriginPro with the following Boltzmann equation:

$$f_{(x)} = A_2 + \frac{A_1 - A_2}{1 + e((X - X_o)/dX)} \tag{2}$$

where $A_2$ = final value, $A_1$ = initial value; $X_o$ = center, and $dX$ = time constant.

The input resistance of mouse DRG neurons was calculated as the slope of individual linear fits of the voltage–current relationships generated from increasing and depolarizing current injection square pulses. $dV/dt$ vs. voltage plots were generated with ClampFit from action potentials elicited by injecting 280 pA. From each individual $dV/dt$ vs. voltage plot, we extracted the action potential properties, namely the resting potential (Vrest), the threshold membrane potential (Vthres), the maximal voltage value (Vmax), the repolarization voltage (Vrepol), and the slopes representing the depolarization and repolarization phases (Supplementary Fig. 7b). Linear fitting was done using OriginPro.

Statistical analyses were performed using GraphPad Instat 3 software and Estimation Stats[58]. Individual tests are described on each of the figure legends. No technical replicates were included in the analyses.

**Reporting summary**. Further information on research design is available in the Nature Research Reporting Summary linked to this article.

## Data availability

Data supporting the findings of this manuscript are available from the corresponding authors upon reasonable request. A reporting summary for this article is available as a Supplementary Information file. The source data underlying figures and Supplementary Figures are provided as a Source Data file, https://doi.org/10.6084/m9.figshare.12192630

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

## Acknowledgements

We thank Dr. G.R. Lewin for providing N2A$^{Piezo1-/-}$ cells, Dr. R.C. Foehring for critically reading the manuscript, and members of the Vásquez, Cordero, and Chesler laboratories for technical support. This work was supported by the National Institutes of Health (R01GM33845 to V.V. and to J.F.C.-M.), the American Heart Association (16SDG26700010 to V.V.), the Intramural program at National Center for Complementary and Integrative Health (NCCIH-IRP to A.T.C.), and the Neuroscience Institute at UTHSC (Postdoctoral Matching Salary Support to R.C.). Additional support was provided by NCATS and funded through the NCATS/DPI Trans-NIH HEAL Initiative (A.T.C.).

## Author contributions

Lead author: V.V. Conceptualization: V.V. and J.F.C.-M. Methodology: V.V., J.F.C.-M., A.T.C., and L.O.R. Formal analysis: V.V. and L.O.R. Investigation: L.O.R., R.C., and A.R. N. Resources: V.V., J.F.C.-M., and A.T.C. Writing—original draft preparation: V.V., J.F. C.-M., and L.O.R. Writing—review and editing: V.V., J.F.C.-M., A.T.C., and L.O.R. Supervision: V.V., J.F.C.-M., and A.T.C. Project administration: V.V. Funding acquisition: V.V., J.F.C.-M., and A.T.C.

## Competing interests

The authors declare no conflict of interest.
