## [Peer Review File · Nature Communications]

Reviewers' Comments:

Reviewer #1:

Remarks to the Author:

"A dietary fatty acid counteracts neuronal mechanical sensitization"

SUMMARY: This manuscript presents a thoughtful and rigorous characterization of the effects of margaric acid (MA) enrichment of the plasma membrane on PIEZO2-dependent mechanotransduction in a variety of mammalian cell types: mouse and rat DRG neurons, N2A mouse neuroblastoma cell line, Merkel cell carcinoma cells, and human iPSC-derived sensory neurons. Prior to this study, MA was shown to inhibit mechanically evoked Piezo1 currents. In this current study, the authors observed that Piezo2 is less sensitive to MA than Piezo1. They also show that Piezo2 but not Piezo1 is perturbed by pharmacological disruption of the actin cytoskeleton, echoing earlier findings with Piezo1. Several of the authors' key experiments, most notably the testing of Piezo1-Piezo2 chimeric channels, suggest that the beam domain of Piezo1 and Piezo2 confers differential dependence of the channels on cytoskeleton and their different sensitivities to MA. Using mouse, rat, and iPSC-derived human DRG neurons, the authors show that MA partially attenuates rapidly adapting currents (attributable to Piezo2) and that MA increases the apparent threshold for mechanically evoked currents in the cell indentation assay. They also demonstrate that MA decreases the ability of sensory neurons to recover from 1Hz indentation stimuli. They propose that MA affects mechanotransduction channels and not voltage-gated sodium or steady-state potassium channels, and show that resting membrane potential is not perturbed with MA treatment. Finally, the authors demonstrate that MA attenuates the effect of bradykinin, an inflammatory endogenous peptide that sensitizes Piezo2 and other cation channels to mechanical stimuli via G protein signaling, on mouse DRG neurons. Overall, this manuscript makes substantial strides to advance our understanding how Piezo1 and Piezo2 sense membrane tension and mechanical force, which is incompletely understood.

The authors' conclusions are generally supported by the data in its present form. We have several suggestions for additional analyses and/or experiments to better support the claims in the paper, in particular with regard to the proposed effects of MA on DRG neurons and the conclusions drawn from the Piezo2 chimera experiments. However, even in the absence of further revision experiments we believe this paper is suitable for publication with text revisions and/or analysis of existing data to address the items detailed below.

Points of concern (note these are all relatively minor)

1. Electrophysiology of mouse DRG neurons (Figure 4 and Supplemental Figure 5):

- a. It is difficult to conclude from the data as-is that MA does not affect DRG neuronal excitability (and hence only affects mechanically-gated ion channels, i.e. Piezo2). For example, not all K⁺ channels modulate resting potential. It would be good to know if the input resistance of these cells is changed after MA, which could be calculated from your existing voltage-clamp experiments. From the methods, it looks like the bath solutions include many cations that could carry currents aside from stretch-activated ion channels.
- b. Furthermore, the AP waveform (4A and Supplemental Figure 5E) does look slightly different in the traces presented, contrary to the authors' conclusions regarding MA and neuronal excitability/AP firing. Specifically, the after hyperpolarization and the shoulder are smaller after MA. This could be due to cell-to-cell variability (i.e. minor differences in the chosen traces). A dv/dt plot, or assessment of other AP waveform parameters in your analysis software could reveal differences due to MA effects on other channels, or convincingly demonstrate no difference. Any differences should be noted and addressed in the main text.
- c. You have some inactivating inward currents after the step from positive potential to resting/negative in Supplemental Figure 5A. Similar to concern 1a, is this observation reproducible over multiple cells and if so could this be due to effects of MA on voltage-dependent ion channels?
- d. The chosen 1Hz stimulus protocol is nice (Figure 4C and elsewhere). Have you observed

differences in recovery from inactivation +/- MA using a voltage-clamp paradigm similar to Coste et al. PNAS (2013), varying Δt and comparing the time constant of recovery? This is by no means an essential experiment.

2. Cytoskeleton disruption experiments with Piezo2, Piezo1, and beam chimera in N2A cells (Figure 2 and Supplemental Figure 3):

a. The conferral of MA sensitivity in 2E and loss of LatA effect in 2F are striking. More specific discussion of how this directly relates to previous experimental findings, such as Cox et al. (2016) should be included.

b. The Eijlkamp et al. (2013) paper on EPAC1 and Piezo2 by no means demonstrates that Piezo2 requires cytoskeletal elements for normal function (line 60), it suggests cytoskeletal involvement but does not prove direct interaction. Treatment with cytoskeletal-disrupting drugs such as colchicine and latrunculin A can have many other effects, for example trafficking and recycling of other membrane proteins. I would tone down this citation. Furthermore, the citation on line 122 cites the above article as well as Coste et al. (2010), Moroni et al. (2018), and Wang et al. (2019) as evidence that Piezo2 requires cytoskeleton. None of these other papers examined that directly.

Reviewer #2:

Remarks to the Author:

The authors illustrate that Piezo2 currents are inhibited by Margarinic Acid (MA) in N2A mouse neuroblastoma cells, rodent DRGs and human iPSC-derived DRGs. This study, a follow up to a previous study from the same group, finds that compared to Piezo1, greater amounts of MA are required for Piezo2 inhibition. However, disruption of the actin cytoskeleton by Latrunculin A reduces the required amount. MA also inhibited action potentials from mouse and rat DRG neurons, and reversed sensitization of mouse DRG currents by bradykinin. The authors suggest that MA could be used to attenuate overactive touch responses as seen in tactile allodynia.

This is a well-executed study, the data are of high quality, and the experiments are appropriately controlled and analyzed.

Major Comments

The authors suggest that the observations on the relationship between Piezo2 activity and MA supplementation suggest a novel method to reduce the effects of overactive sensory receptors in tactile allodynia. However, for translation to human DRGs and the proposed clinical applications, it is important to show the effect of MA on human iPSC-derived neurons electrophysiology in current-clamp mode - does MA treatment inhibit action potentials like in the rodent DRGs? Also, is the effect of MA on bradykinin-sensitized currents (as in Fig. 6) also observed in the human iPSC-derived neurons? These experiments are necessary to support the conclusions and proposed impact of the findings (e.g. "Our findings....suggest new avenues to treat tactile allodynia"). If the authors are unable to perform these experiments, then the claims should be adjusted accordingly throughout the manuscript.

Figure 2 - The authors conclude in this figure that the different beam regions between Piezo1 and Piezo2 is the determinant for their different responses to MA supplementation based on a chimeric Piezo2 channel that possesses a Piezo1 beam. Their argument would be more robust if they included another chimera of Piezo1 with a Piezo2 beam region. If this new chimera under the treatment of MA and Latrunculin A behaved more like Piezo2 in response to MA, the authors would be able strengthen their conclusion that the beam region in each protein is responsible for the individual protein's response to MA. While this experiment would be highly informative in providing a mechanistic basis of the observed effect, in light of the current Covid-19 disruptions and lab closures, it may not be feasible to do these experiments. As such, I leave it to the author's discretion.

Minor Comments

Line 116: "Thus, as with PIEZO1, PIEZO2 is less active in rigid membranes (>78 pN (ref 24))."
It is not clear what >78 pN refers to, nor where this number arises from.

Supplemental Figure 3A- The colors and layout chosen for this figure make it confusing and hard to read. Please consider changing the layout and colors to make it easier to follow.

Line 262 - 264. This speculative statement should be moved to the Discussion section.

REVIEWER COMMENTS:

We thank the reviewers for the time invested in our manuscript, especially during the current global situation, and for their enthusiasm for our work. Thanks to their comments and suggestions, our work has been improved with new analyses, text additions, and a new experimental result. We think these changes have made our manuscript stronger. Please find below a point-by-point response to the specific issues raised by the reviewers. Additions and changes are highlighted in yellow throughout the manuscript file.

Reviewer #1 (Remarks to the Author):

“A dietary fatty acid counteracts neuronal mechanical sensitization”

SUMMARY: This manuscript presents a thoughtful and rigorous characterization of the effects of margaric acid (MA) enrichment of the plasma membrane on PIEZO2-dependent mechanotransduction in a variety of mammalian cell types: mouse and rat DRG neurons, N2A mouse neuroblastoma cell line, Merkel cell carcinoma cells, and human iPSC-derived sensory neurons. Prior to this study, MA was shown to inhibit mechanically evoked Piezo1 currents. In this current study, the authors observed that Piezo2 is less sensitive to MA than Piezo1. They also show that Piezo2 but not Piezo1 is perturbed by pharmacological disruption of the actin cytoskeleton, echoing earlier findings with Piezo1. Several of the authors' key experiments, most notably the testing of Piezo1-Piezo2 chimeric channels, suggest that the beam domain of Piezo1 and Piezo2 confers differential dependence of the channels on cytoskeleton and their different sensitivities to MA. Using mouse, rat, and iPSC-derived human DRG neurons, the authors show that MA partially attenuates rapidly adapting currents (attributable to Piezo2) and that MA increases the apparent threshold for mechanically evoked currents in the cell indentation assay. They also demonstrate that MA decreases the ability of sensory neurons to recover from 1Hz indentation stimuli. They propose that MA affects mechanotransduction channels and not voltage-gated sodium or steady-state potassium channels, and show that resting membrane potential is not perturbed with MA treatment. Finally, the authors demonstrate that MA attenuates the effect of bradykinin, an inflammatory endogenous peptide that sensitizes Piezo2 and other cation channels to mechanical stimuli via G protein signaling, on mouse DRG neurons. Overall, this manuscript makes substantial strides to advance our understanding how Piezo1 and Piezo2 sense membrane tension and mechanical force, which is incompletely understood.

The authors' conclusions are generally supported by the data in its present form. We have several suggestions for additional analyses and/or experiments to better support the claims in the paper, in particular with regard to the proposed effects of MA on DRG neurons and the conclusions drawn from the Piezo2 chimera experiments. However, even in the absence of further revision experiments we believe this paper is suitable for publication with text revisions and/or analysis of existing data to address the items detailed below.

Points of concern (note these are all relatively minor):

1. Electrophysiology of mouse DRG neurons (Figure 4 and Supplemental Figure 5):

a. It is difficult to conclude from the data as-is that MA does not affect DRG neuronal excitability (and hence only affects mechanically-gated ion channels, i.e. Piezo2). For example, not all K⁺ channels modulate resting potential. It would be good to know if the input resistance of these cells is changed after MA, which could be calculated from your existing voltage-clamp experiments. From the methods, it looks like the bath solutions include many cations that could carry currents aside from stretch-activated ion channels.

We thank the reviewer for bringing this up. Based on this suggestion, we have now analyzed our existing current-clamp results and determined the input resistance as analyzed elsewhere² (see figure below). Importantly, the input resistance for margaric acid (MA)-treated dorsal root ganglia (DRG) neurons is not significantly different from the control. This new analysis has been added to Supplementary Figure 6e-f and described accordingly in the Results section (lines 226-232). Nevertheless, we have toned down this statement in the manuscript by explicitly saying that the effect of MA on other ion channels needs to be analyzed on a case-by-case basis.

a) Membrane potential changes elicited by stepwise current injections of control (black) and MA (red; 300 μ M)-treated DRG neurons. $n = 8$. b) Input resistance of control and MA (300 μ M)-treated DRG neurons. Input resistance was calculated as the slope of linear fits of current-voltage plots (shown on a) generated from a series of increasing current injection steps². n is denoted above the x-axis. Mann-Whitney test. n.s. indicates values not significantly different from the control.

b. Furthermore, the AP waveform (4A and Supplemental Figure 5E) does look slightly different in the traces presented, contrary to the authors' conclusions regarding MA and neuronal excitability/AP firing. Specifically, the after hyperpolarization and the shoulder are smaller after MA. This could be due to cell-to-cell variability (i.e. minor differences in the chosen traces). A dv/dt plot, or assessment of other AP waveform parameters in your analysis software could reveal differences due to MA effects on other channels, or convincingly demonstrate no difference. Any differences should be noted and addressed in the main text.

We would like to thank the reviewer for bringing this up and we agree that without this analysis, we could not have convincingly said that MA does not affect the excitability of the DRG neurons. Based on this reviewer's comments, we have performed a phase plot analysis to characterize AP features of control and MA-treated DRG neurons, as analyzed elsewhere³. To that end, we used the APs elicited with current injection in mouse DRG neurons. The phase plot analysis allowed us to extract the resting and threshold membrane potentials (V_{rest} and V_{thres} , respectively), maximal voltage peak (V_{max}), the repolarization potential (V_{repol}), and the depolarization and repolarization slopes from each AP trace. The Figure and Table below show that there are no major differences between the AP properties of the control and MA-treated DRG neurons. This new analysis is now part of the Supplementary Material and is noted in the main text (Supplementary Figure 7, Table I, and lines 226-229).

Mouse DRG neurons

a) Current-clamp recordings of membrane potential changes elicited by current injection in control and MA (300 μ M)-treated DRG neurons. b) Phase plot analysis of action potentials depicted in a. (c-h) Mean differences between control and MA (300 μ M)-treated DRG neurons are shown in the above Gardner-Altman estimation plots¹ for the resting and threshold membrane potentials (V_{rest} and V_{thres} , respectively), maximal voltage peak (V_{max}), the repolarization potential (V_{repol}), and the depolarization and repolarization slopes from each AP trace shown in a. Experimental groups are plotted on the left axes; the mean differences are plotted on floating axes on the right as a bootstrap sampling distribution. The mean differences are depicted as a dot; the 95% confidence interval is indicated by the ends of the vertical error bar. p values were determined with two-sided permutation t -test.

Table I. Mean values \pm SD of V_{rest} , V_{thres} , V_{max} , and V_{repol} , and depolarizing and repolarizing action potential slopes.

AP properties	Control	MA (300 μ M)	p value
Resting membrane potential (V_{rest} , mV)	-61.02 ± 0.69	-62.28 ± 1.88	0.280
Threshold membrane potential (V_{thres} , mV)	-29.12 ± 2.16	-28.89 ± 3.56	0.883
Maximal voltage peak (V_{max} , mV)	55.58 ± 2.67	52.99 ± 3.86	0.155
Repolarization potential (V_{repol} , mV)	-61.45 ± 1.83	-59.88 ± 3.75	0.326
Depolarizing slope (mV/ms)	2.73 ± 1.06	2.53 ± 0.31	0.704
Repolarizing slope (mV/ms)	-0.01 ± 0.32	-0.07 ± 0.06	0.627

p values were determined with two-sided permutation t -test. $n = 7$ for control and MA-treated DRG neurons.

c. You have some inactivating inward currents after the step from positive potential to resting/negative in Supplemental Figure 5A. Similar to concern 1a, is this observation reproducible over multiple cells and if so could this be due to effects of MA on voltage-dependent ion channels?

We thank the reviewer for noticing the differences between the representative records shown on Supplemental Figure 5a. Based on this reviewer's comments, we determined the magnitude of the currents shortly after the square-pulse voltage protocol. The figure below shows that there are no significant differences between the inactivating inward current magnitudes after the experimental pulses of control and MA-treated DRG neurons. This new analysis has been added to Supplementary Figure 5d and described accordingly in the Results section (lines 223-224). Moreover, we have changed the representative traces to avoid misleading the readers.

a) Representative whole-cell patch-clamp recordings of control and MA (300 μ M)-treated mouse DRG neurons depolarized in a stepwise manner from a holding potential of -80 mV. * denotes the time interval selected to determine the current magnitude after the voltage-pulse protocol. b) Peak currents within the first 10 ms interval after the end of the voltage protocol vs. pulse potential from control and MA (300 μ M)-treated mouse DRG neurons. $n = 7$. Unpaired t -test and Mann-Whitney test. n.s. indicates values not significantly different from the control.

d. The chosen 1Hz stimulus protocol is nice (Figure 4C and elsewhere). Have you observed differences in recovery from inactivation +/- MA using a voltage-clamp paradigm similar to Coste et al. PNAS (2013), varying Δt and comparing the time constant of recovery? This is by no means an essential experiment.

We have not varied the Δt in our 1Hz stimulus protocols, but it is something we would definitely like to test in the future. Initially, we did not consider this experiment (before the reviewer's question) because MA does not change inactivation in PIEZO2 when expressed in heterologous systems or endogenously expressed in mouse and human neurons.

We are very grateful for this reviewer's comments and, thanks to the analysis suggestions, we now have more evidence to propose that MA does not affect DRG neuronal excitability. As mentioned above, we have toned down this statement in the manuscript by explicitly saying that the effect of MA on other ion channels needs to be analyzed on a case-by-case basis.

2. Cytoskeleton disruption experiments with Piezo2, Piezo1, and beam chimera in N2A cells (Figure 2 and Supplemental Figure 3):

a. The conferral of MA sensitivity in 2E and loss of LatA effect in 2F are striking. More specific discussion of how this directly relates to previous experimental findings, such as Cox et al. (2016) should be included.

We agree with the reviewer and we have now added the following sentences to the Results and Discussion sections:

Lines 143-145, Results section: Our results further support a previous work that demonstrated that PIEZO1 gating depends on the plasma membrane tension using bleb-attached patches in the absence of the cytoskeleton⁴.

Lines 308-310: For instance, Cox et al. demonstrated that PIEZO1 is solely gated by bilayer tension in the absence of the cytoskeleton⁴, whereas current evidence suggests that PIEZO2 gating might depend on the cytoskeleton⁵.

b. The Eijlkamp et al. (2013) paper on EPAC1 and Piezo2 by no means demonstrates that Piezo2 requires cytoskeletal elements for normal function (line 60), it suggests cytoskeletal involvement but does not prove direct interaction. Treatment with cytoskeletal-disrupting drugs such as colchicine and latrunculin A can have many other effects, for example drugs such as colchicine and latrunculin A can have many other effects, for example trafficking and recycling of other membrane proteins. I would tone down this citation. Furthermore, the citation on line 122 cites the above article as well as Coste et al. (2010), Moroni et al. (2018), and Wang et al. (2019) as evidence that Piezo2 requires cytoskeleton. None of these other papers examined that directly.

We apologize for this oversight. Hence, we toned down the aforementioned citations as follows:

Lines 62-63: Moreover, it has been suggested that PIEZO2 requires cytoskeletal elements such as actin and tubulin for normal function⁵.

Lines 129-132: Although there is no direct evidence that PIEZO2 needs an intact cytoskeleton for gating, previous works have shown that it cannot be gated in excised patches, given the notion that the cytoskeleton is required for activation⁵⁻⁸. On the contrary, PIEZO1 can be solely activated by membrane tension in inside-out patches^{4,9,10}.

Reviewer #2 (Remarks to the Author):

The authors illustrate that Piezo2 currents are inhibited by Margarinic Acid (MA) in N2A mouse neuroblastoma cells, rodent DRGs and human iPSC-derived DRGs. This study, a follow up to a previous study from the same group, finds that compared to Piezo1, greater amounts of MA are required for Piezo2 inhibition. However, disruption of the actin cytoskeleton by Latrunculin A reduces the required amount. MA also inhibited action potentials from mouse and rat DRG neurons, and reversed sensitization of mouse DRG currents by bradykinin. The authors suggest that MA could be used to attenuate overactive touch responses as seen in tactile allodynia. This is a well-executed study, the data are of high quality, and the experiments are appropriately controlled and analyzed.

Major Comments

The authors suggest that the observations on the relationship between Piezo2 activity and MA supplementation suggest a novel method to reduce the effects of overactive sensory receptors in tactile allodynia. However, for translation to human DRGs and the proposed clinical applications, it is important to show the effect of MA on human iPSC-derived neurons electrophysiology in current-clamp mode - does MA treatment inhibit action potentials like in the rodent DRGs?

This is an interesting point, but one we tried and unfortunately were unable to address. Although we got near-complete conversion of the human iPSC-derived neurons to low-threshold mechanoreceptors, these cells take a long time to mature in culture. This means that while the cells all express PIEZO2 and fire action potentials, overall the mechanically-evoked currents and neural excitability were reduced relative to acutely-cultured neurons from rodents. Even though we have previously shown that human iPSC-derived neurons can elicit action potentials by current injection¹¹, we have not been able to elicit action potentials with mechanical stimuli. Accordingly, we have toned down our translation claims throughout the manuscript (highlighted in yellow throughout the manuscript: lines 35-36, 84-86, 269-271, 333-335).

Also, is the effect of MA on bradykinin-sensitized currents (as in Fig. 6) also observed in the human iPSC-derived neurons? These experiments are necessary to support the conclusions and proposed impact of the findings (e.g. “Our findings.....suggest new avenues to treat tactile allodynia”). If the authors are unable to perform these experiments, then the claims should be adjusted accordingly throughout the manuscript.

Unfortunately, we did not perform bradykinin experiments in human iPSC-derived neurons. Accordingly, we will tone down our translation claims throughout the manuscript (highlighted in yellow).

Figure 2 - The authors conclude in this figure that the different beam regions between Piezo1 and Piezo2 is the determinant for their different responses to MA supplementation based on a chimeric Piezo2 channel that possesses a Piezo1 beam. Their argument would be more robust if they included another chimera of Piezo1 with a Piezo2 beam region. If this new chimera under the treatment of MA and Latrunculin A behaved more like Piezo2 in response to MA, the authors would be able strengthen their conclusion that the beam region in each protein is responsible for the individual protein’s response to MA. While this experiment would be highly informative in providing a mechanistic basis of the observed effect, in light of the current Covid-19 disruptions and lab closures, it may not be feasible to do these experiments. As such, I leave it to the author’s discretion.

We agree with the reviewer. We obtained the *Piezo1-Piezo2* beam chimera construct after the initial submission and we were able to record from it before the University's closure (see figure below). As expected, the inhibition by MA of this new chimera is not as efficient as seen for PIEZO1 (panel b, blue bar). Notably, this effect is modulated by latrunculin A treatment (panel c, blue bars), similar to the results observed for PIEZO2. These results support the idea that the PIEZO2 beam is a key region tuning MA-mediated channel inhibition. We believe that these results strengthen our conclusion. We have added these results to Figure 2 and lines 168 and 182 of the Results section accordingly.

a) Top, ribbon representation of PIEZO2 monomer (PDB ID: 6KG7; gray) highlighting the residues that were exchanged for those of PIEZO1 (yellow). Bottom, ribbon representation of PIEZO1 monomer (PDB ID: 5Z10; gray) highlighting the residues that were exchanged for those of PIEZO2 (blue). b) Inhibition by MA (100 μM)-supplementation of N2A cells and N2A^{Piezo1-/-} cells transfected with Piezo2, and Piezo2-Piezo1 and Piezo1-Piezo2 beam chimeras. n is denoted above the x-axis. Unpaired *t*-test and Mann-Whitney test. c) Normalized current densities elicited by maximum displacement of MA (100 μM; 18 h)-supplemented N2A cells (expressing endogenous Piezo1) and N2A^{Piezo1-/-} cells transfected with Piezo2, and Piezo2-Piezo1 and Piezo1-Piezo2 beam chimeras treated with and without latrunculin A. n is denoted above the x-axis. Unpaired *t*-test (for PIEZO1) and Mann-Whitney test (for PIEZO2, and PIEZO2-PIEZO1 and PIEZO1-PIEZO2 beam chimeras). Asterisks indicate values significantly different from control (***p*<0.01 and ****p*<0.001) and n.s. indicates values not significantly different from the control.

Minor Comments

Line 116: “Thus, as with PIEZO1, PIEZO2 is less active in rigid membranes (>78 pN (ref 24)).”24).” It is not clear what >78 pN refers to, nor where this number arises from.

“>78 pN” refers to the mean tether force of plasma membranes enriched in margaric acid that we previously determined using atomic force microscopy¹². For clarity, we have removed this number from the sentence.

Supplemental Figure 3A- The colors and layout chosen for this figure make it confusing and hard to read. Please consider changing the layout and colors to make it easier to follow.

We agree with the reviewer. To avoid confusion, we decided to remove this Supplementary Figure.

Line 262 - 264. This speculative statement should be moved to the Discussion section.

As per reviewer's recommendation, we have removed this statement.

References

- 1 Ho, J., Tumkaya, T., Aryal, S., Choi, H. & Claridge-Chang, A. Moving beyond P values: data analysis with estimation graphics. *Nat Methods* **16**, 565-566, doi:10.1038/s41592-019-0470-3 (2019).
- 2 Yi, F. *et al.* Autism-associated SHANK3 haploinsufficiency causes Ih channelopathy in human neurons. *Science* **352**, aaf2669, doi:10.1126/science.aaf2669 (2016).
- 3 Trombin, F., Gnatkovsky, V. & de Curtis, M. Changes in action potential features during focal seizure discharges in the entorhinal cortex of the in vitro isolated guinea pig brain. *J Neurophysiol* **106**, 1411-1423, doi:10.1152/jn.00207.2011 (2011).
- 4 Cox, C. D. *et al.* Removal of the mechanoprotective influence of the cytoskeleton reveals PIEZO1 is gated by bilayer tension. *Nat Commun* **7**, 10366, doi:10.1038/ncomms10366 (2016).
- 5 Eijkelkamp, N. *et al.* A role for Piezo2 in EPAC1-dependent mechanical allodynia. *Nat Commun* **4**, 1682, doi:10.1038/ncomms2673 (2013).
- 6 Coste, B. *et al.* Piezo1 and Piezo2 are essential components of distinct mechanically activated cation channels. *Science* **330**, 55-60, doi:10.1126/science.1193270 (2010).
- 7 Moroni, M., Servin-Vences, M. R., Fleischer, R., Sanchez-Carranza, O. & Lewin, G. R. Voltage gating of mechanosensitive PIEZO channels. *Nat Commun* **9**, 1096, doi:10.1038/s41467-018-03502-7 (2018).
- 8 Wang, L. *et al.* Structure and mechanogating of the mammalian tactile channel PIEZO2. *Nature* **573**, 225-229, doi:10.1038/s41586-019-1505-8 (2019).
- 9 Lewis, A. H. & Grandl, J. Mechanical sensitivity of Piezo1 ion channels can be tuned by cellular membrane tension. *Elife* **4**, doi:10.7554/eLife.12088 (2015).
- 10 Syeda, R. *et al.* Piezo1 Channels Are Inherently Mechanosensitive. *Cell Rep* **17**, 1739-1746, doi:10.1016/j.celrep.2016.10.033 (2016).
- 11 Nickolls, A. R. *et al.* Transcriptional Programming of Human Mechanosensory Neuron Subtypes from Pluripotent Stem Cells. *Cell Rep* **30**, 932-946 e937, doi:10.1016/j.celrep.2019.12.062 (2020).
- 12 Romero, L. O. *et al.* Dietary fatty acids fine-tune Piezo1 mechanical response. *Nat Commun* **10**, 1200, doi:10.1038/s41467-019-09055-7 (2019).

Reviewers' Comments:

Reviewer #1:

Remarks to the Author:

The authors have made substantial text changes and added new data analyses and results that greatly strengthen the manuscript. They have directly addressed all of our concerns highlighted in our initial review. It is our opinion that the data now fully support the claims, and no further revisions are required.

Reviewer #2:

Remarks to the Author:

The authors have nicely addressed my comments, and further strengthened an already-strong manuscript.

One remaining discrepancy arising from the edits is:

Lines 331 - 333 (in Discussion): "Our data show that MA does not impair bradykinin-mediated sensitization, but rather reduces the mechano-currents similar to the ones measured during normal neuronal mechanical response or non-inflammatory-like levels."

seems to be at odds with:

Lines 84 - 86 (in Introduction): "Importantly, MA decreases PIEZO2 currents potentiated by the proalgesic agent bradykinin, indicating that it might be particularly useful for reducing heightened touch responses during inflammation."

*** Reviewer #2 (Remarks to the Author):**

The authors have nicely addressed my comments, and further strengthened an already-strong manuscript. One remaining discrepancy arising from the edits is:

Lines 331 - 333 (in Discussion): "Our data show that MA does not impair bradykinin-mediated sensitization, but rather reduces the mechano-currents similar to the ones measured during normal neuronal mechanical response or non-inflammatory-like levels." seems to be at odds with:

Lines 84 - 86 (in Introduction): "Importantly, MA decreases PIEZO2 currents potentiated by the proalgesic agent bradykinin, indicating that it might be particularly useful for reducing heightened touch responses during inflammation."

Thanks to this reviewer, we have rewritten this statement as follows:

Our data show that MA is able to counteract PIEZO2 sensitization by bradykinin by reducing the mechano-currents to non-inflammatory-like levels.